# Emerging and Clinically Accepted Biomarkers for Hepatocellular Carcinoma

**DOI:** 10.3390/cancers16081453

**Published:** 2024-04-10

**Authors:** Sami Fares, Chase J. Wehrle, Hanna Hong, Keyue Sun, Chunbao Jiao, Mingyi Zhang, Abby Gross, Erlind Allkushi, Melis Uysal, Suneel Kamath, Wen Wee Ma, Jamak Modaresi Esfeh, Maureen Whitsett Linganna, Mazhar Khalil, Alejandro Pita, Jaekeun Kim, R. Matthew Walsh, Charles Miller, Koji Hashimoto, Andrea Schlegel, David Choon Hyuck Kwon, Federico Aucejo

**Affiliations:** 1Department of Hepato-Pancreato-Biliary & Liver Transplant Surgery, Digestive Diseases and Surgery Institute, Cleveland Clinic Foundation, Cleveland, OH 44195, USA; sxf306@case.edu (S.F.); hongh2@ccf.org (H.H.); sunk2@ccf.org (K.S.); jiaoc@ccf.org (C.J.); zhangm5@ccf.org (M.Z.); grossa8@ccf.org (A.G.); allkuse2@ccf.org (E.A.); uysalm@ccf.org (M.U.); khalilm5@ccf.org (M.K.); pitaa@ccf.org (A.P.); kimj30@ccf.org (J.K.); walshm@ccf.org (R.M.W.); hashimk@ccf.org (K.H.); schlega4@ccf.org (A.S.); kwonc2@ccf.org (D.C.H.K.); 2Department of Hematology and Oncology, Taussig Cancer Institute, Cleveland Clinic Foundation, Cleveland, OH 44195, USA; kamaths@ccf.org (S.K.); maw4@ccf.org (W.W.M.); 3Department of Gastroenterology, Hepatology, and Nutrition, Digestive Diseases and Surgery Institute, Cleveland Clinic Foundation, Cleveland, OH 44195, USA; modarej@ccf.org (J.M.E.); linganm@ccf.org (M.W.L.)

**Keywords:** hepatocellular carcinoma, biomarkers, liver transplant, liver resection, liver cancer

## Abstract

**Simple Summary:**

This review article serves to update physicians on what biomarker tests are available for the surveillance and follow up of hepatocellular carcinoma. Our goal is to provide an update on newly arising tests and their diagnostic accuracy when compared to the standard protocol. Findings from this article hope to promote further investigation into high utility tests and continue the pursuit of clinical utility for emerging biomarkers such as circulating tumor DNA.

**Abstract:**

Hepatocellular carcinoma (HCC) is the third leading cause of cancer-related death and the sixth most diagnosed malignancy worldwide. Serum alpha-fetoprotein (AFP) is the traditional, ubiquitous biomarker for HCC. However, there has been an increasing call for the use of multiple biomarkers to optimize care for these patients. AFP, AFP-L3, and prothrombin induced by vitamin K absence II (DCP) have described clinical utility for HCC, but unfortunately, they also have well established and significant limitations. Circulating tumor DNA (ctDNA), genomic glycosylation, and even totally non-invasive salivary metabolomics and/or micro-RNAS demonstrate great promise for early detection and long-term surveillance, but still require large-scale prospective validation to definitively validate their clinical validity. This review aims to provide an update on clinically available and emerging biomarkers for HCC, focusing on their respective clinical strengths and weaknesses.

## 1. Introduction

Hepatocellular carcinoma (HCC) is the third leading cause of cancer-related death worldwide and the sixth most diagnosed malignancy worldwide [1,2,3]. Liver transplant is the most common curative-intent surgical approach, and HCC accounts for 20–40% of liver transplants performed worldwide [1,2,3]. Biomarkers are objective and quantifiable characteristics of a biological process intended to provide context to a patient’s condition without the use of subjective data [4]. The biomarkers for HCC play a critical role in the management of these complex patients, including diagnosis, prediction of outcomes, long-term follow up, and need for re-intervention [5]. Serum alpha-fetoprotein (AFP) is the traditional, ubiquitous biomarker for HCC [1]. However, as we continue to learn about this disease and the now well-established limitations of AFP, it becomes clearer that the use of multiple biomarkers is essential to optimizing care for these patients [5]. We aim to review the existing literature surrounding the current and emerging biomarkers for HCC to aid both clinicians and researchers in the study and management of HCC.

## 2. Materials and Methods

This study is a narrative review of the literature. In terms of search strategy, the authors began by searching keywords such as “hepatocellular carcinoma”, “hepatocellular carcinoma biomarkers”, “emerging biomarkers”, “hepatocellular carcinoma early diagnosis”, “hepatocellular carcinoma postoperative surveillance”, “hepatocellular carcinoma ctDNA”, and “hepatocellular carcinoma salivary markers” in the National Library of Medicine database PubMed^®^. The relevant articles were compiled and reviewed by the authors. Following this step, the authors investigated selected articles’ Cited by and References sections. Relevant articles in these sections were once again compiled and reviewed, with their data incorporated and referenced in this article.

## 3. Biomarkers in Clinical Practice

### 3.1. Early Detection and Diagnosis

As of 2021, AFP was the only biomarker approved for the screening and diagnosis of HCC [1]. However, its poor predictive value has raised ongoing concerns from the clinical community about its long-term use. For example, AFP as a standalone tool demonstrates sensitivity between 39% and 64% and a specificity between 76% and 97% for the new diagnosis of HCC [6]. The initial guidelines for those with cirrhosis recommended a screening abdominal ultrasound every six months, which has an overall sensitivity of 84% and early detection sensitivity of 47% (Figure 1) [1,7,8]. However, as the use of biomarkers has advanced, there has been substantial evidence that a combination of AFP and ultrasonography provides better early detection, with a sensitivity of 63% and specificity of 84% for the combined approach [1]. Trending AFP values can also improve predictive values in early detection versus isolated values at a single time point [1]. This has been reflected in a 2021 study by Tayob et al. that externally validated an early detection screening algorithm (HES) which included AFP level, rate of change of AFP, age, level of alanine aminotransferase, and platelet count [9]. While this screening model resulted in a specificity > 90%, it still demonstrated a sensitivity of just 52.5%, meaning nearly half of negative results are false negatives [9].

For patients with low- to intermediate-level AFP levels (20–200 ng/mL) or AFP-negative HCC, other known and clinically used biomarkers can be utilized in both screening and surveillance. These include prothrombin induced by vitamin K absence II (DCP) and lens culinaris agglutinin-reactive glycosylated form of AFP (AFP-L3) [1]. DCP is an abnormal form of prothrombin that results from defective post-translational modification of the prothrombin precursor protein [5]. The data are mixed comparing DCP to AFP alone as a screening tool. Marrero et al. reported that DCP provided improved diagnostic capability in distinguishing between HCC and cirrhosis versus AFP with a sensitivity of 89% vs. 77% and specificity of 95% vs. 73% [11]. Nakamura et al. found that AFP outperformed DCP when the tumor diameter was <3 cm, but DCP outperformed AFP when the tumor diameter was >5 cm [12]. Most recently, Choi et al. found that DCP did not increase discriminatory power when combined with AFP and AFP-L3% for early HCC detection [13].

AFP-L3 alone is relatively insensitive in the diagnosis of HCC when AFP levels are low; however, a fractionated approach to calculating AFP-L3% did demonstrate a specificity as high as 94% [1,5,14]. Unfortunately, the sensitivity of AFP-L3% is reported at just 37%, much lower than AFP in the comparison cohort [1,15]. Overall, the best clinical use for AFP-L3% and DCP is to add utility in patients with intermediate AFP levels (20–200 ng/mL), as their overall sensitivity in a screening context remains limited [14].

### 3.2. Post-Transplant Prognosis

Serum AFP levels have been described as the tumor characteristic most strongly predictive of post-transplant survival [16]. A retrospective study by Berry et al. showed that patients with serum AFP ≤ 15 ng/mL at the time of transplantation had improved post-transplant mortality [adjusted HR (AHR) = 1.02, 95% CI 0.93–1.12] vs. serum AFP 16–65 ng/mL (AHR = 1.38, 95% CI 1.23–1.54), 66–320 ng/mL (AHR = 1.65, 95% CI = 1.45–1.88), and >320 ng/mL (AHR = 2.37, 95% CI = 2.06–2.73) [16]. This included patients with high tumor burden outside of Milan criteria (one lesion greater ≥2 cm and ≤5 cm, or up to three lesions, each ≥1 cm and ≤3 cm) which would traditionally have precluded transplantation [16,17]. These patients had excellent post-transplant survival if serum AFP level was 0 to 15 ng/mL (AH = 0.97, 95% CI = 0.66–1.43) despite being considered “high-risk” by traditional criteria [16]. Comparatively, patients with excellent Milan criteria but an AFP level greater than or equal to 66 ng/mL demonstrated poor survival (AHR = 1.93, 95% CI = 1.74–2.15) [16]. Shimamura et al. found similar results in their 965-patient retrospective review, which produced a new expanded LT criteria [18]. The 5-5-500 rule study found that for patients with a nodule size ≤ 5 cm in diameter, nodule number ≤ 5, and alfa-fetoprotein value ≤ 500 ng/mL, there was a 5-year recurrence rate of 7.3% (95% CI) with a 19% increase in the number of eligible patients for live transplant [18].

Hameed et al. reported a significant association with vascular invasion when AFP levels surpassed 300 ng/mL and found that an AFP level > 1000 ng/mL was the greatest predictor of vascular invasion and the only significant predictor of tumor recurrence [17,19]. The 5-year recurrence-free survival rate for patients with AFP > 1000 ng/mL was 52.7%, compared to 80.3% in those with AFP ≤ 1000 ng/mL within their study [19]. Duvoux et al. produced similar findings with three subcategories, including AFP < 100 ng/mL at listing, ≥100 ng/mL and <1000 ng/mL, and ≥1000 ng/mL [20]. The 5-year recurrence rate in these groups were 16.2%, 26.8%, and 53%, respectively, and the 5-year overall survival rate was 67.5%, 51.1%, and 39.1%, respectively [20].

Despite the current data available, a consensus on the optimal pre-transplant AFP level to predict post-transplant survival is yet to be determined, and many proposed scores for transplant candidacy have proposed different AFP cutoffs based on tumor size and number [18,21,22,23]. Future large-scale comparative studies are needed to identify the true optimal cut-off that still allows for maximal candidacy.

### 3.3. Post-Operative Surveillance

The Risk Estimation of Tumor Recurrence After Transplant (RETREAT) score [Table 1] is an externally validated score developed to stratify and identify patients who may require future therapies following liver transplantation for HCC [10]. A total of 1016 patients were included in Mehta et al.’s study which developed the RETREAT score incorporating three variables: explant tumor burden, microvascular invasion, and AFP level before transplant [10]. The calculations are detailed in DCP [10]. Scores range from 0 to 8. Patients with completely necrotic tumor on explant after liver transplantation, no microvascular invasion on explant, and an AFP level lower than 20 ng/mL at liver transplant have a RETREAT score of 0 [10]. A score of 0 predicts 1- and 5-year recurrence risks of only 1.0% (95% CI, 0.0–2.1%) and 2.9% (95% CI, 0.0–5.6%), respectively [10]. A patient with a RETREAT score of 5 or higher has predicted 1- and 5-year recurrence risks of 39.3% (95% CI, 25.5–50.5%) and 75.2% (95% CI, 56.7–85.8%), respectively [10]. These results were then compared to a validation cohort with a C statistic of 0.82 (95% CI 0.77–0.86). This study was further validated in a retrospective review utilizing the UNOS database [24]. For HCC recurrence prediction, RETREAT performed well in the study cohort with a C-index of 0.75 (95% CI 0.71–0.79). Additionally, RETREAT performed well regardless of center volume with a C-index of 0.77 (95% CI 0.71–0.83) for centers that performed 40 or fewer LTs for HCC during the study period and 0.73 (95% CI 0.68–0.78) for centers that performed more than 40 LTs [24].

Finally, van Hooff et al. further externally validated the RETREAT score in a European population [25]. They included 203 patients who underwent liver transplant at a single European center, reporting that overall cumulative HCC recurrence rates were 10.6%, 21.3%, and 23.0% at 2, 5, and 10 years, with most recurrences extrahepatic and at multiple sites [25]. Higher RETREAT scores were associated with higher recurrence rates, with a 10-year recurrence rate of 60.5% in patients with RETREAT ≥ 3 (*n* = 65), compared to 6.2% in those with RETREAT ≤ 2 (*n* = 138; *p* < 0.001) [25]. The study concluded that lower RETREAT scores did in fact predict decreased rates of HCC recurrence in European populations as well.

## 4. Emerging Biomarkers

### 4.1. Risk Stratification

There have been a number of single nucleotide polymorphisms (SNPs) associated with an increased risk of developing HCC, leading to their proposed use in risk stratification and screening for HCC. Previously, genome wide association studies identified nine genes with a role in HCC development, including Intron 1 of *TPTE2* (OR 0.27 [0.19–0.39]), *KIF1B* (OR 0.61 [0.55–0.67]), *GRIK1* (OR 0.84 [0.80–0.89]), *STAT4* (OR 1.22 [1.15–1.29]), *HLA-DQA1/DRB1* (OR 1.28 [1.22–1.35]), *MICA* (OR 1.39 [1.27–1.52]), *HLA-DQ* (OR 1.51 [1.38–1.66]), *DEPDC5* (OR 1.75 [1.51–2.03]), and the upstream of *DDX18* (OR 3.38 [2.07–5.53]) [5,26,27,28,29,30,31]. Of note, the odds ratios of most of these genes are less than 1.5, the accepted cutoff for clinical utility [26].

More recently, Wang et al. highlighted five SNPs involving genes *PNPLA3* and *SAMM50,* all with OR > 1.5, believed to be highly associated with the development of HCC after controlling for both BMI and hepatitis virus infection [32]. Both genes are located on chromosome 22q13.31 and *PNPLA3* has previously been strongly associated with the development of HCC and NAFLD [33,34,35,36].

### 4.2. Early Detection and Diagnosis

#### Totally Non-Invasive Markers: Saliva and Breath

Salivary and breath biomarkers offer a non-invasive alternative to early detection and diagnostic efforts for HCC. Hershberger et al. tested the metabolites of 125 patients (43 healthy, 37 HCC, and 30 cirrhosis) and were able to identify four metabolites with a substantial difference across groups (acetophenone, octadecanol, lauric acid, 3-hydroxybutyric acid) [37,38]. Four tree-based machine-learning models were subsequently developed utilizing twelve total salivary metabolites (octadecanol, acetophenone, lauric acid, 1-monopalmitin, dodecanol, salicylaldehyde, glycyl-proline, 1-monostearin, creatinine, glutamine, serine, and 4-hydroxybutyric acid) with an emphasis on the four previously mentioned metabolites. These models were then trained on a 99-patient cohort and validated on an 11-patient cohort. Their iRF4 model, which placed the most emphasis on the four specific metabolites, had the highest sensitivity (87.9%) and specificity (95.5%) for diagnosis of HCC across all models (Figure 2) [38].

MicroRNA (miRNA) levels have also been detected in the saliva of patients with HCC, offering another potential salivary biomarker [39]. miRNAs are circulating non-coding RNA strands hypothesized to contribute to tumor oncogenesis and progression [40]. Mariam et al. performed small RNA sequencing on 39 patients (20 HCC, 19 cirrhosis) and detected a total of 4565 precursor and mature miRNAs [39]. A predictive model, optimized to the 10 miRNAs they found to be most common between salivary samples and tissue samples (hsa-mir-576, hsa-miR-576-5p, hsa-mir-6727, hsa-mir-27b, hsa-miR-27b-3p, hsa-mir-4664, hsa-mir-125a, hsa-miR-6727-5p, hsa-mir-190b, and hsa-miR-125a-5p), was developed and trained on patients with HCC and chronic liver disease vs. cirrhosis [39]. Without including covariates (age, sex, race, BMI, and smoking) the model was found to have a sensitivity of 80% and specificity of 74% for differentiating HCC from cirrhosis, whereas when covariates were included, the model was found to have a sensitivity of 100% and specificity of 84%.

Studies have further analyzed breath metabolomics in HCC. Specifically, Miller-Atkins et al. reported a predictive model analyzing the breath of 292 patients for volatile organic compounds (VOCs) which included (E)-2-Nonene, ethane, benzene, methylhexane, decene, acetaldehyde, pentane, acetone, isoprene, trimethyl amine, dimethyl sulfide, and more [41]. (E)-2-Nonene, ethane, and benzene were found to be most different between healthy patients and patients with HCC, while acetone, acetaldehyde, and dimethyl sulfide were found to be most different in patients with cirrhosis vs. HCC [41]. Three machine-learning models were created (age and sex, metabolites only, and all three variables). The three-variable model was the superior model, accurately identifying 67 of the 92 patients with HCC, resulting in an overall classification accuracy of 72% [41]. The sensitivity and specificity for HCC were 73% and 71%, respectively [41].

**Figure 2 cancers-16-01453-f002:**
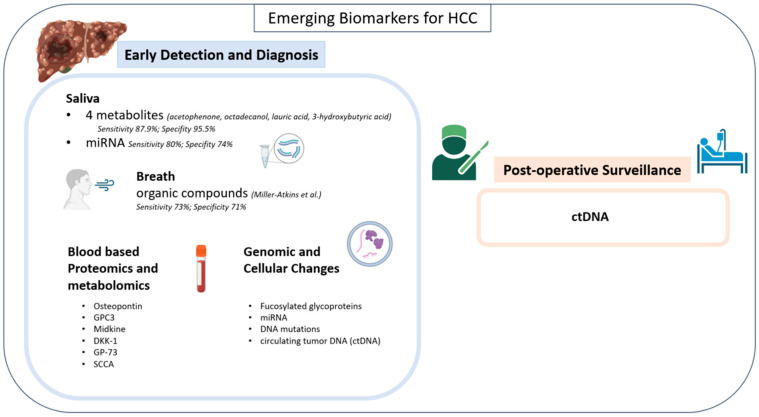
Emerging biomarkers for HCC [41].

### 4.3. Blood-Based Proteomics/Metabolomics

Osteopontin is an integrin-binding phosphoprotein that mediates cell signaling and is involved in regulating tumor progression in bone and epithelial cells [1,5]. It has also been linked to a high rate of vascular involvement of HCC [5]. Shang et al. compared osteopontin to AFP for the differentiation of early HCC from cirrhosis and found it to be superior, with a sensitivity of 75% and a specificity of 62% [42]. When combined, a sensitivity of 83% and specificity of 63% were achieved [42].

Glypican-3 (GPC3) is a plasma membrane-bound heparan sulfate proteoglycan that regulates cell growth by modulating tyrosine kinase activity and the Wnt signaling pathway [5]. Xu et al. compared GPC3 to AFP for the diagnosis of HCC and found an increased sensitivity of 57% compared to 52% for AFP and found a sensitivity of 77% when combined [5,43].

Midkine is a heparin-binding growth factor involved in cell growth, invasion, and angiogenesis during malignant transformation [44]. Elevated levels were detected in very early-stage AFP negative NASH-related HCC. In Vongsuvanh et al.’s AFP-negative HCC cohort, 59.18% (*n* = 29/49) had elevated MDK; the study identified an optimal cut-off of 0.44 ng/mL. Using AFP ≥ 20 IU/mL or MDK ≥ 0.44 ng/mL, a significantly greater number (76.7%; *n* = 66/86) of HCC cases were detected [44]. Within this cohort, Midkine testing had a sensitivity of 68.8% and specificity of 100%.

Dickkopf-1 (DKK-1) is a secretory antagonist of the Wnt signaling pathway investigated by Jang et al. for its use as an HCC biomarker in combination with AFP levels. DKK-1 alone had a sensitivity of 50.0% and specificity of 80.8% for HCC diagnosis [45]. When combined with AFP, DKK-1 resulted in superior diagnostic value when compared to osteopontin, with a sensitivity of 78.4% and specificity of 72.5% [45].

Golgi protein-73 (GP-73), a transmembrane protein located within the Golgi complex, is commonly elevated in chronic liver diseases and substantially elevated in HCC. GP73 was found to be superior to AFP in differentiating HCC from cirrhosis, but not superior to AFP for early HCC diagnosis [46,47].

Squamous cell carcinoma antigen (SCCA) is a serine protease inhibitor that is present in squamous epithelium [40]. Both SCCA and the SCCA-IgM immune complex have been investigated for a potential role in the early detection and diagnosis of HCC. SCCA was found to have a high sensitivity for HCC at 89% but a poor specificity when differentiating from cirrhosis at 50% [48,49].

Overall, each of these offers great additive potential in the management of HCC, especially when combined with imaging and AFP, yet the majority of these have not been prospectively validated, which is necessary for widespread clinical implementation.

### 4.4. Genomic and Cellular Changes

Serologic protein level changes are not the only investigational biomarkers for HCC. Blood-based genomic and cellular changes have also been described for diagnostic use for HCC.

Fucosylated glycoproteins have perhaps been the best studied in this context [50,51,52,53,54]. Increased levels of fucosylated hemopexin, fetuin A, alpha 1 antitrypsin, ceruloplasmin, serum paraoxoanase 1, and histidine rich-glycoprotein have all been observed in patients with HCC and correlated with increased fucosylation in HCC biopsied tissue [50,52,55,56,57,58]. Perhaps the two most well-described alterations are fucosylated kininogen and glycosylated haptoglobin [59,60]. The addition of serum fucosylated kininogen allowed Wang M et al. to increase the early detection rate of AFP negative (<20 ng/mL) HCC from 0% to 89% [59]. When combining the use of AFP levels with testing for levels of both total haptoglobin and specifically the alpha-2,6-sialylation and/or alpha-1,6-fucosylation forms, Ang et al. were able to achieve a sensitivity of 79% and specificity of 95% [60]. Like proteomics/metabolomics, these genomic alterations require large-scale prospective validation to achieve widespread clinical utility.

In addition to the aforementioned epigenetic alterations, miRNA, DNA mutations, and circulating tumor DNA (ctDNA) biomarkers have been described (Figure 2).

Zhang et al. have proposed four specific miRNAs as biomarkers, including miR-16-2-3p, 92a-3p, 107, and 3126-5p [61]. Among those four, a 3-miRNA panel including 92a-3p, 107, and 3126-5p was used to test patients with HCC of all stages. When analyzed for area under curve (AUC), the 3-miRNA panel resulted in an AUC value of 0.969, much improved versus AFP alone (AUC = 0.816) [61]. The combination of AFP and the 3-miRNA panel resulted in an AUC of 0.994, a promising result for this phase II study.

ctDNA, or cell-free DNA (cfDNA), is the most recent emerging biomarker in the detection and management of HCC [62]. ctDNA allows for identification of oncogenic mutations, observation of tumor dynamics, and detection of disease or residual tumor tissue post-treatment [63]. Oussalah et al. found that in 289 patients with cirrhosis, 98 of whom had HCC, ctDNA testing for the *SEPT9* promoter site methylation displayed a sensitivity of 94.1% and a specificity of 84.4% in distinguishing HCC from cirrhotic patients [64]. Iizuka et al. found that their ctDNA assay for *GSTP1* levels had a sensitivity of 69.2% and a specificity of 93.3% in discriminating HCC and HCV carriers at the optimal cut-off value of 73.0 ng/mL [65]. Finally, Wen et al. found that their panel, which tested for four hypermethylated CpG island markers (RGS10, ST8SIA6, RUNX2, and VIM), had the ability to differentiate between 36 HCC patients and control subjects, which included 17 cirrhosis patients and 38 healthy individuals, with a sensitivity of 94% and a specificity of 89% [66].

Wang J et al. compared ctDNA peripheral blood levels to AFP peripheral blood levels in 81 patients with HCC prior to undergoing hepatectomy [67]. In total, 57 of the 81 patients (70.4%) had detectable ctDNA levels prior to surgery, while AFP was only positive in 46 patients (56.8%), indicating a potentially superior diagnostic capability [67]. Positive preoperative ctDNA status was related to larger tumor size, multiple tumor lesions, microvascular invasion, and shorter disease-free survival and overall survival [67].

It is important to distinguish tumor-informed vs. tumor-naïve ctDNA. Tumor-informed ctDNA compares blood-based mutations with a mutational analysis of sampled tumor tissue, while tumor-naïve ctDNA reports the most common mutations as pre-determined by the manufacturer [68,69]. Chan et al. compared tumor-informed versus tumor-naïve ctDNA for the early detection of colorectal cancer (CRC). DNA isolated from tumor tissues was sequenced and a total of 61 somatic genomic alterations in 10 genes were identified [68]. The pre-operative ctDNA detection rate was significantly higher in stage I–III CRC patients using the tumor-informed approach compared to the tumor-agnostic approach, with detection rates of 66% and 31%, respectively (*p*-value = 0.008) [68]. There is not yet a similar study published in the HCC literature.

### 4.5. Post-Operative Surveillance

ctDNA has shown significant promise for the surveillance of HCC [70]. Raj et al. conducted a prospective comprehensive database study in which 96 patients with advanced unresectable HCC received immunotherapy + locoregional therapy (LRT) [70]. ctDNA was assessed for each patient at the discretion of the treating physician and obtained using the Guardant360 platform, which assesses mutations over 500 genes to establish a genomic footprint for the calculation of tumor mutational burden (TMB) [70]. Of the ninety-six patients, eleven showed complete response to immunotherapy based on mRECIST scoring, four of whom ultimately underwent curative-intent resection. ctDNA was cleared in three of these four patients, indicating their progression from TMB+ to TMB- over the course of their treatment. This development indicated that ctDNA can be an objective measurement of response to treatment for HCC.

Of the 57 patients with positive pre-operative ctDNA levels in Wang et al.’s study, 53 had available ctDNA testing post-operatively. Tests found no ctDNA levels in 23 patients (43.4%), a decrease in the mutant allele frequency (MAF) of ctDNA in 13 patients (24.5%), and either an increase of MAF or novel mutations in the ctDNA of 17 patients (32.1%) [67]. Patients were then divided into an increase cohort and a decrease/ctDNA negative cohort. In the increase cohort, 16 of the 17 patients (94.1%) had recurrence of HCC, while only 15 of the 36 (41.7%) of the decreased/negative cohort had recurrence of HCC [67].

Similar studies have been published discussing the promise for ctDNA in long-term surveillance for CRC [68]. Along with their comparison of tumor-informed vs. tumor-naïve ctDNA, Chan et al. compared the utility of landmark ctDNA (defined as the detection of ctDNA from the first plasma sample drawn after the completion of definitive treatment) vs. longitudinal ctDNA in 31 patients with long-term follow up. A total of twenty-nine of the thirty-one patients had available landmark ctDNA levels, seven of whom (24%) had detectable ctDNA levels [68]. The recurrence rate was significantly higher for ctDNA-positive patients at 57% (4/7), compared to 9% (2/22) for negative patients (*p* < 0.05), with a sensitivity and specificity for detection of recurrence of 67% and 87%, respectively [68]. Plasma samples were then evaluated for all thirty-one patients with long-term follow-up via three to four samples following the conclusion of treatment. A total of ten patients tested positive during the longitudinal segment of the study, an increase from the original seven with a positive landmark test [68]. Six of the ten patients (60%) developed CRC recurrence, while zero of the twenty-one with negative longitudinal results developed CRC recurrence, thus increasing the sensitivity to 100% (*p* < 0.05) [68]. There is, again, not yet a similar study published in the HCC literature.

## 5. Discussion

Current approaches to early detection and post-operative surveillance rely heavily on AFP as the sole biomarker of HCC [1,6,7,8,9,16,17]. Unfortunately, we demonstrate that there remains significant diagnostic uncertainty utilizing AFP as a biomarker. When compared to AFP, ctDNA, salivary metabolites, and salivary miRNA models all demonstrated potential for substantial superiority in diagnosis, early detection, and post-operative surveillance in the case of ctDNA [37,38,39,41,64,65,66]. However, these biomarkers are yet to be studied in large-scale prospective models and lack years of clinical utilization compared to AFP.

Ultimately, an understanding of the preliminary test data can guide clinicians to utilize these emerging tests most effectively. Salivary miRNA with covariates provides a high sensitivity non-invasive test to clinicians with the potential to replace other modalities, such as AFP and US, as the predominant screening test for HCC [1,3,6,7,37,38,39,41]. Positive results would motivate clinicians to pursue more invasive or costly confirmatory studies.

There is substantial promise in these emerging biomarkers that should motivate researchers to continue data collection as well as promote the refinement of existing tests. Clinicians should be motivated to pursue potential algorithms that incorporate these emerging biomarkers as data continue to be uncovered.

The limitations of this study include an inherent delay in data, as published works are likely months to years behind current research and data. Additionally, data for the review were collected via a search of keywords in the National Library of Medicine followed by an investigation into cited works as well as articles referencing the specific publication. As a result, data collection for this study was dependent on search engine results and the data collection of other studies.

## 6. Conclusions

Alpha-fetoprotein, AFP-L3, and DCP have described clinical utility for HCC, but unfortunately, they also have well established and significant limitations. Further investigation into effective biomarkers for HCC remains critical for early detection and long-term follow up of patients post-treatment. Circulating tumor DNA, genomic glycosylation, and even totally non-invasive salivary biomarkers demonstrate great promise, but still require large-scale prospective validation to definitively validate their clinical validity. A comparison table has been included with referenced sensitivities and specificities (Table 2).

## Figures and Tables

**Figure 1 cancers-16-01453-f001:**
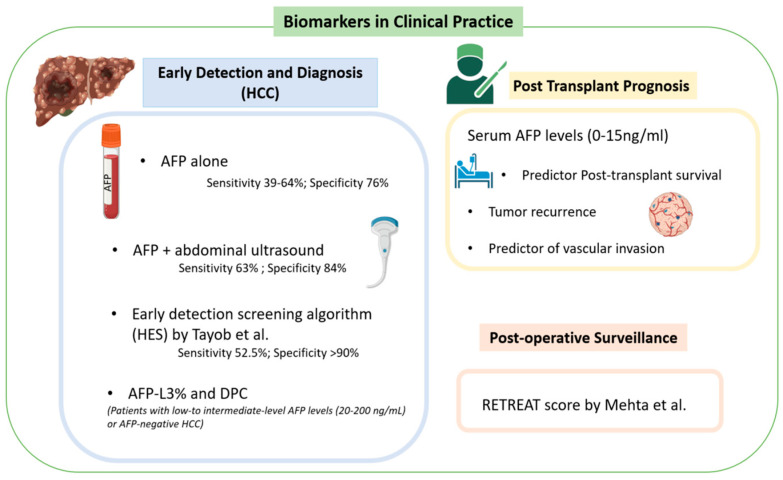
Biomarkers for HCC in clinical practice [1,9,10].

**Table 1 cancers-16-01453-t001:** The Risk Estimation of Tumor Recurrence After Transplant (RETREAT) score calculation [10].

Variable	Points Assigned
AFP level at Time of Liver Transplant	
0–20 ng/mL	0
20–99 ng/mL	1
100–999 ng/mL	2
≥1000 ng/mL	3
Explant Pathology	
Microvascular invasion absent	0
Microvascular invasion present	2
Tumor Size (Explant largest diameter of viable tumor + number of viable tumors)	
0 cm	0
1.1–4.9 cm	1
5–9.9 cm	2
≥10 cm	3

**Table 2 cancers-16-01453-t002:** Sensitivities and specificities mentioned throughout the review are organized below.

Biomarker	Sensitivity	Specificity
AFP alone [6]	39–64%	76–97%
Ultrasound (US) q6mo [1]	84%	
AFP and US q6mo [1]	63%	84%
HCC early detection screening algorithm (HES) [9]	52.5%	>90%
Prothrombin induced by vitamin K absence II (DCP) [11]	89%	95%
Lens culinaris agglutinin-reactive glycosylated form of AFP % (AFP-L3%) [5]	37%	94%
12 salivary metabolites model [38]	87.9%	95.5%
Breath volatile organic compounds (VOCs) [41]	73%	71%
Salivary miRNA without covariates [39]	84%	74%
Salivary miRNA with covariates [39]	100%	84%
Osteopontin [42]	75%	62%
Osteopontin and AFP [42]	83%	63%
Glypican-3 (GPC3) [43]	57%	
GPC3 and AFP [43]	77%	
Midkine [44]	68.8%	100%
Dickkopf-1 (DKK-1) [45]	50%	80.8%
DKK-1 and AFP [45]	78.4%	72.5%
Squamous cell carcinoma antigen (SCCA) [48]	89%	50%
Glycosylated haptoglobin and AFP [60]	79%	95%
ctDNA: *SEPT9* promoter site methylation [64]	94.1%	84.4%
ctDNA: *GSTP1* [65]	69.2%	93.3%
ctDNA: hypermethylated CpG island markers (RGS10, ST8SIA6, RUNX2, and VIM) [66]	94%	89%

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
