# Peer review of "Emerging and Clinically Accepted Biomarkers for Hepatocellular Carcinoma"

_cancers, 2024, doi:10.3390/cancers16081453_

Round 1
Reviewer 1 Report
Comments and Suggestions for Authors
* The authors aimed to provide an update on clinically-available and emerging biomarkers for HCC, focusing on their respective clinical strengths and weaknesses.
* The topic is important and addresses an important gap in the current literature as Hepatocellular Carcinoma (HCC) is the third leading cause of cancer related death and the sixth most diagnosed malignancy worldwide.
* The article is overall well written, but lacks a discussion section in which the authors should show their critical point of view on the topics of the article. Pleas add such section that will help readers to better understand your findings.
* English Language are fine.
Author Response
Reviewer 1
* The authors aimed to provide an update on clinically-available and emerging biomarkers for HCC, focusing on their respective clinical strengths and weaknesses.
* The topic is important and addresses an important gap in the current literature as Hepatocellular Carcinoma (HCC) is the third leading cause of cancer related death and the sixth most diagnosed malignancy worldwide.
* The article is overall well written, but lacks a discussion section in which the authors should show their critical point of view on the topics of the article. Pleas add such section that will help readers to better understand your findings.
* English Language are fine.
Thank you for your feedback. The above recommendations have all been implemented in the review. We agree the discussion was significantly additive to the manuscript!
Reviewer 2 Report
Comments and Suggestions for Authors
Dear authors,
Hepatocellular carcinoma is an important cause of morbidity and mortality, and every effort must be made to detect the condition early and to follow the patient's progress. I think that your study has a certain value and deserves to be published after a revision of certain elements. Thus, I would have some recommendations and suggestions.
Abstract. -please define DCP in the abstract
I think that in the legend of Figure 1 you should include the references cited in the figure.
I think a chapter on material and methods would be necessary, namely a flowchart that outlines the method of searching and selecting the articles included in this review.
Also, a chapter in which to discuss and summarize the results obtained, as well as to analyze the limitations or opportunities opened by your study.
Author Response
Reviewer 2
Hepatocellular carcinoma is an important cause of morbidity and mortality, and every effort must be made to detect the condition early and to follow the patient's progress. I think that your study has a certain value and deserves to be published after a revision of certain elements. Thus, I would have some recommendations and suggestions.
Thank you for your time and review!
Abstract. -please define DCP in the abstract
This has been added!
I think that in the legend of Figure 1 you should include the references cited in the figure.
The Figure itself was unable to be edited but the citations were included in the figure caption
I think a chapter on material and methods would be necessary, namely a flowchart that outlines the method of searching and selecting the articles included in this review.
A chapter on materials and methods was included on page 2
Also, a chapter in which to discuss and summarize the results obtained, as well as to analyze the limitations or opportunities opened by your study.
Thank you for this feedback, another reviewer also suggested this. A discussions chapter was included on page 9 which summarized the opportunities opened, critical results, and limitations of our study

Reviewer 3 Report
Comments and Suggestions for Authors
This review presents the literature on emerging biomarkers for hepatocellular carcinoma, replacing the traditionally used biomarkers AFP, AFP-L3 and DCP.
The results of SNPs, miRNAs, proteomics, metabolomics and genomic markers used for early diagnosis and post-operative surveillance of hepatocellular carcinoma are comprehensively presented.
A very interesting review, but a few suggestions.
â‘ The paper format is ‘Review’.
â‘¡ Page 2, line 13. Tayob et. Al⇒et.al   Check also after.
â‘¢ Page 3, line 19. 66 ⇒ 66 ng/mL
â‘£ Page 3, line 7 from the bottom.
 Table 2 number to Table 1, change insertion position.
⑤ 2.3 Post-operative Surveillance
 This section is AFP-only literature, but please provide examples of DCP reporting.
â‘¥ Page 7, line 14.
Change the number in Table 1 to Teble 2 and insert at the end of the sentence.
Author Response
Reviewer 3
This review presents the literature on emerging biomarkers for hepatocellular carcinoma, replacing the traditionally used biomarkers AFP, AFP-L3 and DCP.
The results of SNPs, miRNAs, proteomics, metabolomics and genomic markers used for early diagnosis and post-operative surveillance of hepatocellular carcinoma are comprehensively presented.
Thank you for your kind words and time in reviewing this manuscript.
A very interesting review, but a few suggestions.
â‘ The paper format is ‘Review’.
The format has been edited
â‘¡ Page 2, line 13. Tayob et. Al⇒et.al   Check also after.
Page 2 et. Al and all other et Al were changed to et. Al
â‘¢ Page 3, line 19. 66 ⇒ 66 ng/mL
Page 3 this suggestion was implemented
â‘£ Page 3, line 7 from the bottom.
 Table 2 number to Table 1, change insertion position.
Page 4 and 5 as well as 9 and 10 the tables were appropriately rearranged and relabeled
⑤ 2.3 Post-operative Surveillance
 This section is AFP-only literature, but please provide examples of DCP reporting.
Unfortunately this data was not found in the literature over the course of writing the article and not again upon re-review. In other words, it seems this data has not yet been published but we agree that it would be additive if available.
â‘¥ Page 7, line 14.
Change the number in Table 1 to Table 2 and insert at the end of the sentence.
Page 4 and 5 as well as 9 and 10 the tables were appropriately rearranged and relabeled.
Round 2
Reviewer 2 Report
Comments and Suggestions for Authors
Dear authors,
Thank you for your responses. The manuscript has been improved; but please check once again references for Figure 1 - I think the correct ones are 9 and 23.
Author Response
Thank you. You are correct that reference 23 (now numbered 10) should be included. Reference 1 should also, so references are now 1, 9, and 10 (10 is former 23).